# The Effects of Heatwaves on Human Morbidity in Primary Care Settings: A Case-Crossover Study

**DOI:** 10.3390/ijerph19020832

**Published:** 2022-01-12

**Authors:** Mahmoud Alsaiqali, Katrien De Troeyer, Lidia Casas, Rafiq Hamdi, Christel Faes, Gijs Van Pottelbergh

**Affiliations:** 1Epidemiology and Social Medicine (ESOC), University of Antwerp, 2610 Antwerp, Belgium; lidia.casasruiz@uantwerpen.be; 2Department of Public Health and Primary Care, KU Leuven, 3000 Leuven, Belgium; katrien.detroeyer@kuleuven.be (K.D.T.); gijs.vanpottelbergh@kuleuven.be (G.V.P.); 3Royal Meteorological Institute of Belgium, B-1180 Brussels, Belgium; rafiq.hamdi@meteo.be; 4Data Science Institute (DSI), I-BioStat, Hasselt University, BE-3500 Hasselt, Belgium; christel.faes@uhasselt.be

**Keywords:** heatwaves, primary care, case-crossover

## Abstract

Purpose: This study assesses the potential acute effects of heatwaves on human morbidities in primary care settings. Methods: We performed a time-stratified case-crossover study to assess the acute effects of heatwaves on selected morbidities in primary care settings in Flanders, Belgium, between 2000 and 2015. We used conditional logistic regression models. We assessed the effect of heatwaves on the day of the event (lag 0) and X days earlier (lags 1 to X). The associations are presented as Incidence Density Ratios (IDR). Results: We included 22,344 events. Heatwaves are associated with increased heat-related morbidities such as heat stroke IDR 3.93 [2.94–5.26] at lag 0, dehydration IDR 3.93 [2.94–5.26] at lag 1, and orthostatic hypotension IDR 2.06 [1.37–3.10] at lag 1. For cardiovascular morbidities studied, there was only an increased risk of stroke at lag 3 IDR 1.45 [1.04–2.03]. There is no significant association with myocardial ischemia/infarction or arrhythmia. Heatwaves are associated with decreased respiratory infection risk. The IDR for upper respiratory infections is 0.82 [0.78–0.87] lag 1 and lower respiratory infections (LRI) is 0.82 [0.74–0.91] at lag 1. There was no significant effect modification by age or premorbid chronic disease (diabetes, hypertesnsion). Conclusion: Heatwaves are associated with increased heat-related morbidities and decreased respiratory infection risk. The study of heatwaves’ effects in primary care settings helps evaluate the impact of heatwaves on the general population. Primary care settings might be not suitable to study acute life-threatening morbidities.

## 1. Introduction

According to the Intergovernmental Panel on Climate Change (IPCC)’s Fifth Assessment Report (AR5), global warming is unequivocal. The last 30 years were probably the warmest of the last 1400 years in the Northern Hemisphere, and the temperature is expected to rise further over the 21st century. According to the same report, heatwaves, which are prolonged periods of extremely high temperature, are expected to be more intense, occur more often, and last longer [1]. 

Most previous studies showed that heatwaves are associated with increased overall morbidity. Results for the effect of heatwaves on specific morbidities showed inconsistencies [2]. For example, Turner et al. [3] showed that heatwaves are associated with increased ambulance attendance for cardiovascular and respiratory diseases, especially in the elderly population. Meanwhile, Wang et al. [4] found an increase in emergency hospital admissions (EHAs) from non-external causes (not caused by out-of-human body causes) and renal diseases but not cardiovascular or respiratory diseases. The effect of heatwaves was more pronounced in the vulnerable population such as children and the elderly [3,4,5,6,7], people with chronic diseases [4,8,9], and people with low socioeconomic status [10,11]. 

To date, most research on the effect of heatwaves on morbidity is based on hospital and emergency care data, and research using primary care data are limited [12,13,14,15]. During the heatwave of summer 2013 in England, the number of general practice (GP) consultations increased for heat illnesses compared with similar periods in non-heatwave years (2012, 2014). No increase was detected for asthma, breathing difficulties, cerebrovascular accidents, or cardiovascular symptoms consultations [12,13]. A case-crossover analysis of the effect of ambient temperature on heat-related morbidities among type-2 diabetics showed that increased temperature is associated with increased GP consultation [15]. Primary care is the first contact point for individuals with a health problem and it directly addresses most of their healthcare needs throughout their life. Therefore, studies using data from primary healthcare are needed to cover a wider population and morbidities including mild diseases and symptoms.

In this study, we investigated the short-term effects of heatwaves on the incidence of selected morbidities among patients attending primary care in Flanders (Belgium) between the years 2000 and 2015. We also studied to what extent the effect is modified by age and premorbid chronic medical conditions such as diabetes and hypertension. We hypothesized that heatwaves are associated with an increased incidence of selected acute morbidities in primary care (detailed in the Methods section) and that increased age, and having premorbid medical conditions increase that risk. 

## 2. Materials and Methods

### 2.1. Study Design and Population

This study aimed to assess the occurrence relation between heatwaves and selected acute morbidities from the primary care settings. To achieve that, acute outcomes were studied as a function of previous exposure (heatwaves) using a case-crossover study design.

The crossover study design is used to assess the short-term effects of transient changes in exposure on outcomes [16]. Briefly, in this design, the exposure in the period immediately before the event (the case or hazard periods) is compared with “the usual” exposure at other nearby periods (control or referent periods) within the participant him/herself. In the case of positive association, it is predicted that exposure is more frequent in the hazard period compared with the referent period. 

As both, the case and controls, are from the same patient, this control for the time-invariant characteristics such as age, sex, and socioeconomic status. In addition, as the controls are chosen from the same risk set of cases, the resulting odds ratio (OR) can be used as an unbiased estimator of incidence density ratio (IDR) [17,18]. 

We used the time stratified method to choose the referent days for each case day [19]. In this method, the selected referent days are in the same month and day of the week (DOW) of the case day; this controls the effect of the DOW, month, and year. On average, there were 3–4 referent days for each case day to compare with. 

We included all patients who had GP consultations for preselected morbidities between 1 January 2000 and 31 December 2015 in the INTEGO dataset. INTEGO is a general practice-based morbidity registration network managed by the Academic Center for General Practice at the University of Leuven. The database contains diagnoses, laboratory results, and medication prescriptions for patients who attended selected general practitioners all around Flanders, Belgium. INTEGO is representative of the Flemish population and uses ICPC2 and ICD10 coding systems to classify morbidities. The database is discussed in detail elsewhere [20]. 

### 2.2. Data Collection

#### 2.2.1. Outcome

We selected several morbidities that we presumed could be affected by short-term changes in temperature. These outcomes were chosen based on a literature review combined with an expert opinion. We also created a combined outcome category for each group of morbidities selected. Table 1 shows the list of morbidities included in our analyses.

#### 2.2.2. Exposure

The mean annual temperature in Flanders is 10 °C and varies significantly by region. The mean temperature of the warmest month is <22 °C [21]. We obtained the ambient temperature data from the Royal Meteorological Institute of Belgium [22] for the Uccle meteorological station near Brussels, Belgium, from 2000 to 2015. We used a dichotomic variable to identify heatwaves using the official Belgian definition for a heatwave: A maximum temperature above 25 °C for at least 5 consecutive days, including at least 3 days with temperatures of 30 °C or higher. The severity of heatwaves is described in terms of weight and intensity. Weight is the sum of excess daily average temperature over 20 °C for all days within the heatwave, and intensity is the weight divided by the number of heatwave days. We also included the highest temperature recorded in the different heatwaves.

### 2.3. Data Analysis

The effect of heat waves can be immediate or delayed. To account for that, we studied different lag periods between the event time (selected morbidities) and the exposure (heatwaves). Previous research used mostly short lag periods [2]. We chose lag 0 (exposure on the same day) and lag 1 up to lag 3. 

In our analysis, we worked event by event and not by the semi-expanded format. We created a stratum for each event and compared the exposure on the case and referent days for the same patient. 

As the observations within each stratum are not independent by default, we used conditional logistic regression to estimate our coefficients. A conditional logistic regression estimates the coefficient of exposure conditional on the matching strata and allows for the elimination of the constant related to each stratum [17]. In our model, the heatwave (binary variable) is the independent exposure variable, and the different morbidities are the outcomes; we created a separate model for each outcome. For the combined outcomes, a case day is defined as having any of the morbidities within that group, while the control is defined as having none of them. Only strata with variable exposure will contribute to the estimation of the coefficient. Strata with the case and selected referent days all having the same exposure (either heatwave or non-heatwave days) will not contribute to the estimation of the coefficients. The resulting incidence density ratio is presented along with the 95% confidence interval. We also presented the results for the different lag periods. We choose a *p*-value of <0.05 as a cutoff point for statistical significance. 

We assessed the effect modification by age by including an interaction term between the exposure variable (heatwaves) and the age at diagnosis. We dichotomized the variable age at diagnosis and used 70 years as a cut-off. We also assessed the effect modification by having chronic diseases by including an interaction term between the exposure variable (heatwaves) and having a registered diagnosis of diabetes (DM) and hypertension (HTN) separately. 

The statistical analysis was performed using statistical software (R version 3.5.0) (R Foundation for Statistical Computing, Vienna, Austria) [23]. 

## 3. Results

Throughout the study period of 15 years, we had 1,484,478 consultations with the selected outcomes previously mentioned. Out of these, 22,344 consultations had a variable exposure in their strata, i.e., a heatwave in one day and a non-heatwave in other days. Figure 1 shows the age and sex distribution of our population.

During the study period, there were seven heatwaves. The heatwaves were different in the number of days, intensity, and maximum temperature. Table 2 shows a summary of these heatwaves. 

Appendix A showed the average number of events per day in the heatwave and non-heatwave days for the different morbidities studied. 

Figure 2 shows the IDRs with a 95% confidence interval (CI) for the incidences of general and heat-related morbidities. Heatwaves were, generally, associated with an increased incidence of heat-related morbidities. The IDR of the adverse effect physical factor (A88), which is related to heatstroke, was 3.93 [2.94–5.26] at lag 0, for Dehydration (T11) it was 3.93 [2.94–5.26] at lag 1, and for Orthostatic hypotension (K88) it was 2.06 [1.37–3.10] at lag 1. There was no significant effect on more general morbidities such as feeling ill (A05) and fainting/syncope (A06). The combined estimate for the group was 1.29 [1.13–1.47] at lag 1.

Figure 3 shows the IDRs with 95% CI for the incidences of cardiovascular morbidities. There was no significant effect, except for Transient cerebral ischaemia (K89) and Stroke/cerebrovascular accident (K90). With the increasing lag period between heatwaves and the morbidity, the estimate for TIA was decreasing 0.56 [0.31–1.01] at lag 3, and the estimate for stroke was increasing 1.45 [1.04–2.03] at lag 3.

Figure 4 shows the IDRs with 95% CI for the incidences of respiratory morbidities. Heatwaves were associated with fewer respiratory infections. The IDR of upper respiratory infections (URI) was 0.82 [0.78–0.87] at lag 1, for lower respiratory infections (LRI), it was 0.82 [0.74–0.91] at lag 1, and for pneumonia (R80), it was 0.79 [0.67–0.92] at lag 2. For Shortness of breath/dyspnoea (R02), the heatwaves were associated with an increased risk of 1.49 [1.02–2.17] at lag 3. We did not see a significant effect on asthma (R96) or chronic obstructive pulmonary disease (COPD) (R95). 

The results showed no significant interaction between age or having a chronic disease (DM or HTN) and the heatwaves. Appendix A show the result for age and chronic diseases modification. Appendix A shows the exact results for primary analysis. 

## 4. Discussion

We studied the effect of heat waves on selected morbidities within primary care settings. Heatwaves were associated with an increased incidence of heat-related morbidities such as heat stroke and dehydration. Heatwaves were associated with a decreased incidence of respiratory infections. For the cardiovascular group, it was only significant for Transient cerebral ischaemia (K89) and Stroke/cerebrovascular accident (K90). There was no significant interaction by age or premorbid chronic medical condition (DM, HTN).

Our findings of increased heat-related morbidities in heatwaves are logical and in line with previous research [8,24,25,26]. For example, during the 2006 heatwaves in California, hospitalizations for heat-related illnesses increased by 10 times [8]. This supports the validity of our method as we see the result where we expect to see it. 

For the cardiovascular group, we noted that, with increasing lag, the risk of TIA is decreasing, and the risk of stroke is increasing. This might be explained by the fact the neurological symptoms should remain for a certain period (>24 h) to be diagnosed as a stroke. Our finding is supported by previous research. For example, a case-crossover study showed that extreme high temperature over the last 3 days is associated with increased risk for both ischemic (OR = 1.18; 95% CI: 1.07–1.36) and hemorrhagic strokes (OR = 1.34; 95% CI: 1.26–1.42) [27]. A recent review article concluded that, despite some inconsistencies, there is supporting evidence that temperature is associated with increased stroke risk with the hot temperature tends to have a more immediate effect (i.e., short lag between the exposure and the event) [28].

For the other cardiovascular morbidities studied, we did not see a significant effect. Previous literature showed inconsistent results [3,4,9,29]. In our study, this might be explained by the fact that patients with more serious diseases might head to the emergency department directly and, as a result, are missing in the primary care dataset. 

In the respiratory group, our results showed that heatwaves were associated with a lower incidence of respiratory infections. In general, respiratory infections show clear seasonality and are usually related to cold weather. This might be related to the effect of heat on viral replication and transmission. Previous research showed that respiratory viruses’ replication and transmission are dependent on relative humidity and temperature; namely, it is more present in cold dry conditions. Air transmission efficiency decreased with increased temperature to reach an undetected level at 30 °C [30,31,32,33]. This might explain the reduced incidence of respiratory infection during heatwaves seen in our results. In addition, the patients may decide to not visit their physicians, especially if the symptoms are mild. For asthma and COPD, there was no significant increase during heatwaves. The previous literature showed inconsistent results [34,35,36]. In our study, this might be explained by the fact that there is no specific code for asthma or COPD exacerbation. Patients coming for a regular check-up and medication refill, as well as exacerbation, are registered with the same code in the dataset. 

We did not find a significant interaction by age or having chronic diseases (DM, HTN). The majority of previous research showed that the elderly are more susceptible to the effect of heatwaves [4,5,6,37,38]. The result of the effect modification by premorbid chronic conditions showed inconsistent results [24,39,40]. Our findings could be related to the fact that we studied the effect in the primary care settings not the emergency settings and the population characteristics differ from the ones studied in previous research. This might be a reassuring finding that vulnerable populations receive the heatwave warnings and take the appropriate precautions to protect themselves during heatwaves. 

Our study has several strengths. First, we have conducted our research in primary care settings. The previous research concentrated mainly on urgent care. Primary health care covers a wider range of populations and morbidities, especially mild diseases and symptoms that do not warrant hospital admission. This helps capture the full effect of heatwaves and plan for relevant interventions. Secondly, in our analysis, we worked event by event. Most previous research worked in semi-expanded format under the assumption of rare events; if an event happened in one day, it will not occur in the other days of the month for the same patient. Under that assumption, they can create one stratum for all events that happened on one day and assign a weight to it equal to the number of events on that day without checking whether the subject had the event on the referent days or not. This is true for mortality, as it is impossible to happen twice, and possibly for a limited number of morbidities that are rare. However, this is not true for all morbidities as the subject could have the event more than once in the same month.

In our study, there were several potential limitations. In our research, we used the official definition of a heatwave in Belgium based on the temperature data from one meteorological station (Uccle station). There is a temperature variation between cities and between urban and rural areas in Belgium [41], which might affect the health impact estimate [42]. In further research, different definitions of heatwaves and data at a finer level could be used to study the effect more closely and avoid misclassification. Second, the patients with more severe morbidities might go directly to the hospital without consulting their physicians first. It is not always reported in the dataset, and there are possible inconsistencies in adding the exact date. Third, in our study, we control by design for all the time-invariant factors, and seasonal trends, limiting the possibility of confounding. However, our result could be affected by factors related to daily changes in patient behavior (drinking water, activities, etc.) and air pollution concentrations. Lastly, we have run our analysis for different morbidities and subgroups, thus part of our result might be affected by multiple testing and might not be generalizable or reducible. 

## 5. Conclusions

We studied the effect of heatwaves on the occurrence of human morbidities within primary care settings. Heatwaves are associated with increased heat-related morbidities such as heat stroke, dehydration, and orthostatic hypotension. Heatwaves are associated with decreased respiratory infection risk, as well. For the cardiovascular morbidities studied, we did not find a significant effect except for the increased risk of stroke at lag 3. We did not find a significant effect modification by age or premorbid chronic disease (DM, HTN). 

Our primary work showed that, within primary care settings, heatwaves were associated with an increased incidence of selected morbidities. There was no significant association with acute life-threatening morbidities such as myocardial infarction or arrhythmias. Therefore, data from primary care settings, along with data from emergency care settings, could be used to evaluate the full impact of heatwaves on the general population and the effectiveness of preventive measures. For acute life-threatening morbidities, it would be more suitable to study using hospital or emergency service data. 

Further research should study heat effects at a finer level (zip codes), including possible interactions with other environmental factors such as air pollution. In addition, it should account for human behaviors that could modify the effect of environmental factors on human health. 

## Figures and Tables

**Figure 1 ijerph-19-00832-f001:**
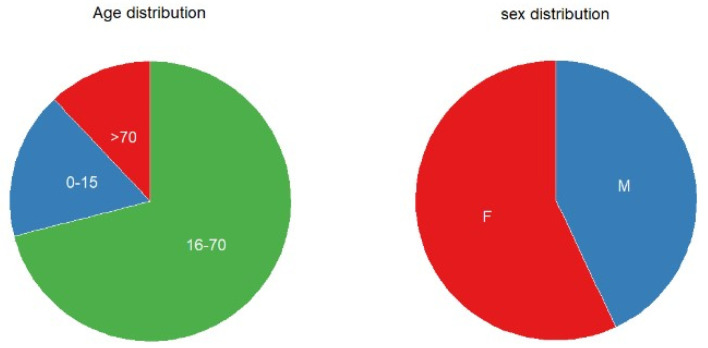
Distribution of the study sample according to age and gender.

**Figure 2 ijerph-19-00832-f002:**
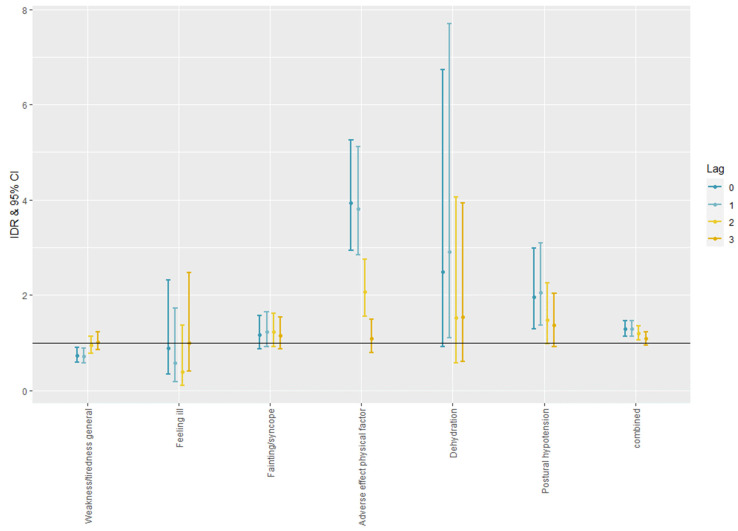
The IDRs with 95% CI for general and heat-related morbidities.

**Figure 3 ijerph-19-00832-f003:**
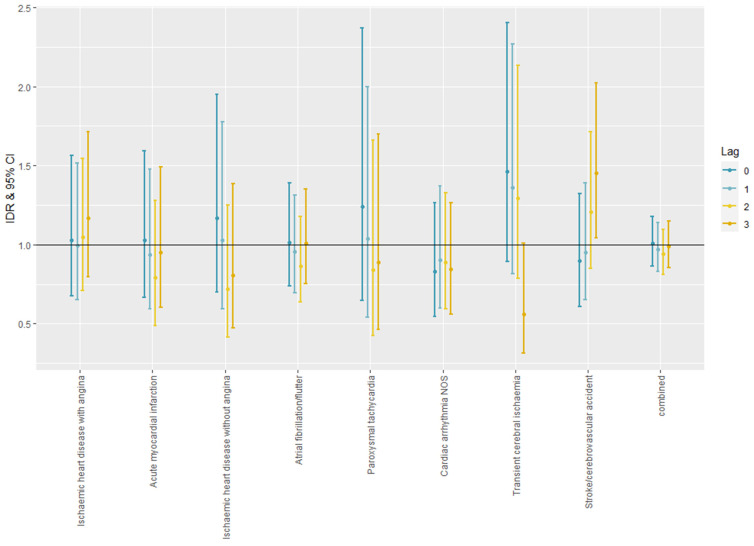
The IDRs with 95% CI for cardiovascular morbidities.

**Figure 4 ijerph-19-00832-f004:**
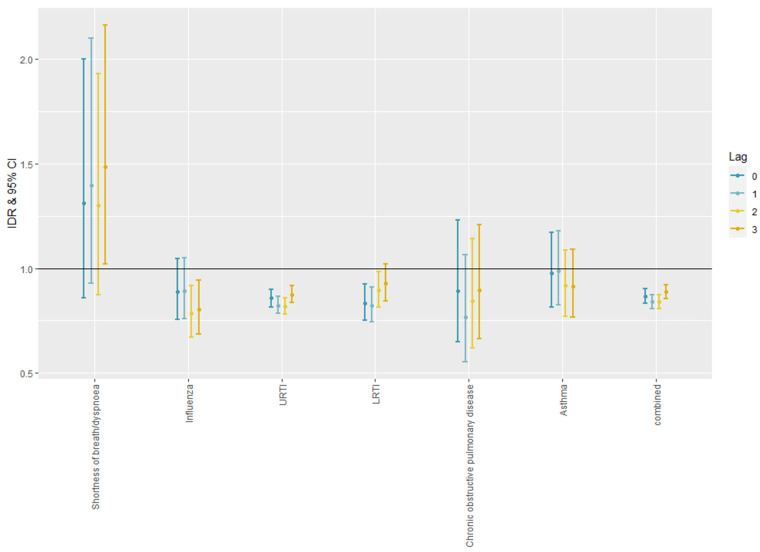
The IDRs with 95% CI for respiratory morbidities.

**Table 1 ijerph-19-00832-t001:** List of morbidities studied along with their corresponding ICPCC2 code.

ICPCC	Diagnosis	ICPCC	Diagnosis
**General and heat-related**		**Respiratory**	
A04	Weakness/tiredness general	R02	Shortness of breath/dyspnoea
A05	Feeling ill	URTI (R74:77)	Upper respiratory infections
A06	Fainting/syncope	LRTI (R78, R81)	Lower respiratory infections
A88	Adverse effect physical factor	R80	Influenza
T11	Dehydration	R95	Chronic obstructive pulmonary disease
K88	Postural hypotension	R96	Asthma
**Cardiovascular**			
K74	Ischaemic heart disease with angina		
K75	Acute myocardial infarction		
K76	Ischaemic heart disease without angina		
K78	Atrial fibrillation/flutter		
K79	Paroxysmal tachycardia		
K80	Cardiac arrhythmia NOS		
K89	Transient cerebral ischaemia (TIA)		
K90	Stroke/cerebrovascular accident		

Note: ICPCC = International Classification of Primary Care code.

**Table 2 ijerph-19-00832-t002:** Summary of heatwaves in Belgium from 2000–2015.

Start	End	Duration	Weight ^a^	Intensity ^b^	Highest Temp ^c^
1-August-2003	13-August-2003	13	52.38	4.03	33.8
18-June-2005	25-June-2005	8	29.97	3.75	32.9
9-June-2006	13-June-2006	5	13.71	2.74	31.3
15-July-2006	30-July-2006	16	68.03	4.25	36.2
7-July-2010	14-July-2010	8	24.61	3.08	33.2
21-July-2013	27-July-2013	7	24.61	3.52	32.4
30-June-2015	5-July-2015	6	28.47	4.74	34.5

^a^ The sum of excess daily average temperature over 20 °C for all days within the heatwave. ^b^ The weight divided by the number of heatwaves days. ^c^ The highest temperature recorded in the heatwaves.

## Data Availability

Data stored at KU Leuven University and are available upon request.

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
