# Peer review of "The Effects of Heatwaves on Human Morbidity in Primary Care Settings: A Case-Crossover Study"

_ijerph, 2022, doi:10.3390/ijerph19020832_

Round 1

Reviewer 1 Report

Good morning, Congratulations and thanks for the effort developing your article. Please reconsider the following: Line2: The title: consider: The Effects of Heatwaves on Human Morbidity in Primary Care Settings in Belgium (2000-2025). Line 13: Abstract: opening statement: Human influence on the climate system is clear, and recent anthropogenic emissions of greenhouse gases are the highest in history. Recent climate changes have had widespread impacts on human and natural systems. The actual STUDY AIM to evaluate the potential ... Line 14: Methods: Line18: Results: Line26: Conclusion:  Line38: consider description of the climate of the study region and its geography: The Flanders region climate is maritime temperate, with significant precipitation in all seasons; the average temperature is 3 °C (37 °F) in January, and 21 °C (70 °F) in July; the average precipitation is 65 millimetres (2.6 in) in January, and 78 millimetres (3.1 in) in July). Line 68: 2. Aim of the Study: This study aimed to assess the occurrence relation between heatwaves and selected acute morbidities from the primary care settings. Line 68: 3.Material and Methods: L74-L83: consider rephrasing. Line 94: consider removal of the (The database is discussed in detail elsewhere) Line 104: Table List of morbidities (find a different and better table design) Line 117: more details needed for lag 1 up to lag 3. Line 132: (IDR) Line 142: Details about Flanders, Belgium climate ... Average year temperature and geographic factors. Line 145: Distribution of the study sample according to age and gender: Line 151: 4.2: Summary of heatwaves in Belgium from 2000-2015:  Line 162: Table needs a better format. Line 164: CI?  Line 168: TIA? L169: Table (choose another table design presentation)  L172-L178: rephrase please. L180: reconsider table design L192-L193: rephrase please L198: TIA abbreviation? L204: rephrase please (no significance at: P=) L208: remove (ED) L215-L216: rephrase  L239: ((he previous researches concentrated on urgent care settings and more research is needed in primary care settings). consider removal L255:  ((The physician usually reports that in the dataset afterward.)) unclear, consider removal. L268-269: explain please L269-272: statistical significance at p=? ... rephrase and explain L278: Recommendations: 

Because your study article is emphasizing heatwaves as part of a global warming phenomena the reader would like to acquire more information regarding level of pollution (Ozone level data) wind factors, and other elements that can have influence on heat generation. Better data quality about the sample households, crowding indices, presence of air conditioning, job related activities, electromagnetic fields ,,, etc. May be mortality data distribution might help understanding the heatwaves effects.

I wish you a good continuity and hope this review will help you develop a more consistent article. I will be happy to review your article again.

best regards

Author Response

Good day, 

Thank you very much for taking the time and effort to review the article. I really appreciate your comments. Please find below our responses for the comments: 

  • For the title, We think that relation between heat and morbidity would be general not limited to the area. Thus we choose to make it general while providing details regarding source of data in the methods section. 
  • Abstract: we were limited by word count(around 200) 
  • We added more details about the weather of Flanders in the exposure section
  • We added more details about the lag: the principle of using it and references support the different lag used  
  • I agree with the remarks about environmental and human factors that could affect our results. We added theses suggestion to our recommendations:  Further research should consider studying environmental factors at finer level(zip codes). Also, it should account for human behaviors that could modify the effect of environmental factor on human health.
  • I updated the draft taking into consideration the other remarks provided

Again, thank you for your time and valuable remarks. I uploaded the updated version. Please let me know if you have other questions or comments.

Sincerely,

Mahmoud

Reviewer 2 Report

Thank you for the opportunity to review this paper. This manuscript investigated the relationship between heatwaves and primary care presentations in Flanders, Belgium between 2000 and 2015. It was shown that while heatwaves present risk factors in some health conditions (e.g. heat stroke, dehydration, stroke) there are also some possible protective factors (e.g. respiratory infections). Whilst the more sever morbidities may not present themselves in primary care data. 

The authors did a good job of presenting their methods, data, acknowledging its limitations, and reflecting with reasoned discussion as to what the authors found rather than inferring the results may be something they are not. 

The paper is well written, fits well within the journals scope and helps inform the the wider public health literature. I have some comments and critiques for the authors to consider in improving this manuscript, that they can decide if are necessary.

  • The medical abbreviation DM, HTN only ever appears throughout the manuscript in its abbreviated form. 
  • I am curious as to why 70y was chosen as the cut off when typically 65y would classify older persons...?
  • Further to my previous point even though you saw no effect for age when analysing the their population, if sub-group analysis by age-groups was done based on expected risk to heat - e.g. very young 0-4, adolescent 5-17, adult 18-39, middle age 40-64, older persons >64, you might see age specific incidence and risk to age specific health conditions that are currently masked as one large group.
  • Expand on potential future work. In the last sentence before the conclusion, you say the purpose was to "generate hypotheses rather than test them". Therefore there should be a few sentences on where this work can be expanded upon next. 

Author Response

Good day, 

Thank you very much for taking the time and effort to review the article. I really appreciate your comments. Please find below our responses for the comments: 

  • For the age: First, we studied the cutoff at 65, we did not find significant interaction. We tried older age to find out if that change the results which did not.
  • We added more suggestions to our recommendations to expand more on potential points for further research

 Again, thank you for your time and valuable remarks. I uploaded the updated version. Please let me know if you have other questions or comments.

Sincerely,

Mahmoud

Reviewer 3 Report

The study used a time-stratified case-crossover and conditional logistic regression models to assess the acute effects of heatwaves at different lags on a few morbidities selected in Belgium. The authors suggested that heatwaves were associated with increased heat-related morbidities and decreased respiratory infection risk. I have major considerations in the methods section as well as the presentation of results.

  1. Heatwave definition. These authors used “a maximum temperature above 25 ° C or at least 5 consecutive days, including at least 3 days with temperatures of 30 °C or higher.” to define the heatwaves. I would suggest authors justifying the definition. Perhaps a sensitive analysis based on different definition is needed in the study.
  2. Lags periods. “we studied different lag periods between the event time and the exposure to capture the possible lag periods between the exposure (heatwaves) and the effect; lag 0 (exposure on the same day), lag 1 up to lag 3” Authors need to provide a reason why lag 1 – lag 3 were selected in the model. Have the authors conducted a statistical test for the lags? Any biological mechanism for the lags? I would suggest authors adding more discussion or justify the selection.
  3. Health data in the primary care setting. During the heatwave the patients aged may have more severe morbidities and they are likely to directly visit hospital. A sensitive analysis for old population is needed.
  4. Data analysis and modelling. It is unclear to me for how these authors used the case-crossover study design and conduct a goodness-of-fit of modelling. More detailed description is needed in the method section. Other daily weather data such as relative or absolutely humidity and environmental factors would be included in the model in assessing the effect of heatwave. These data can be collected.
  5. A descriptive or summary table/figure for the health outcome is needed. It will help readers understand the data distribution.

Author Response

Good day, 

Thank you very much for taking the time and effort to review the article. I really appreciate your comments. Please find below our responses for the comments: 

  • Heatwave definition: we used official definition of heatwaves in Belgium. Previous research showed that the definition of heat wave affect the results. Thus, we decided to choose the official one for our research.
  • Lags periods: we added more details regarding the principle of lag and references to support choosing lag 1-3
  • Old age and morbidity: that was indeed one of the points in our research. We studied effect modification by age in our research. We did not find significant interaction between age and heatwaves
  • Study design and statistical analysis: I updated the methods section to expand more on our study design. This was indeed one of the challenging points in our paper. We tried to balance expanding more on the methods and keeping the paper about the primary research: heatwave and morbidity rather than a method paper. We added references regarding our methods for more information. We also considering writing a method paper to explain all details regarding study design and statistical analysis.
  • The effect of other environmental variable and human factors on the relation between heatwaves and morbidity: I do agree with that. Different environmental factors and human behaviour could affect the relation between heat and morbidity. This was one of the limitation of our research. We added in our recommendations that further research should consider studying heat effect at a finer level(zip codes) including possible interaction with other environmental factors such as air pollution. Also, it should account for human behaviors that could modify the effect of environmental factors on human health.
  • We added summary table for the health outcome in our supplementary material.

Again, thank you for your time and valuable remarks. I uploaded the updated version. Please let me know if you have other questions or comments.

Sincerely,

Mahmoud

Round 2

Reviewer 1 Report

good morning and thank you for providing a revised version and your letter comments.
I believe the manuscript has been improved. The study design is not perfect and the figures need to be differently presented as to allow a proper visualization of the data. 

Best Regards

Author Response

Good day, 

Thank you for your comments. For the data visualization, this was indeed one of the challenging points in our paper. We tried different visualizations till we reach our current graphs. Also, based on the recommendations of the academic editor, we provided the exact results as supplements for further reference and use. 

Sincerely, 

Mahmoud